# Less is More: Improving Molecular Force Fields with Minimal Temporal Information

## Abstract

Accurate prediction of energy and forces for 3D molecular systems is one of fundamental challenges at the core of AI for Science applications. Many powerful and data-efficient neural networks predict molecular energies and forces from single atomic configurations. However, one crucial aspect of the data generation process is rarely considered while learning these models i.e. Molecular Dynamics (MD) simulation. Molecular Dynamics (MD) simulations generate time-ordered trajectories of atomic positions that fluctuate in energy and explore regions of the potential energy surface (e.g., under standard NVE/NVT ensembles), rather than being constructed to steadily lower the potential energy toward a minimum as in geometry relaxations. This work explores a novel way to leverage molecular dynamics (MD) data, when available, to improve the performance of such predictors. We introduce a novel training strategy called FRAMES, that use an auxiliary loss function for exploiting the temporal relationships within MD trajectories. Counterintuitively, on two atomistic benchmarks and a synthetic system we observe that minimal temporal information, captured by pairs of just two consecutive frames, is often sufficient to obtain the best performance, while adding longer trajectory sequences can introduce redundancy and degrade performance. On the widely used MD17 and ISO17 benchmarks, FRAMES significantly outperforms its Equiformer baseline, achieving highly competitive results in both energy and force accuracy. Our work not only presents a novel training strategy which improves the accuracy of the model, but also provides evidence that for distilling physical priors of atomic systems, more temporal data is not always better.

## 1 Introduction

Predicting the quantum properties of atomic systems underpins many tasks in computational chemistry and materials science, yet traditional simulation methods (e.g. ab initio calculations) are often too expensive for large-scale or high-throughput applications. In response, machine learning methods—especially Graph Neural Networks (GNNs) (Wu et al., 2020)—have emerged as a fast and accurate alternative for estimating energies, forces, and other properties, with successful extensions to protein structure prediction, virtual drug screening, and materials design. GNNs generally model atoms as nodes and the physical interaction of two atoms with edges, and also the interaction of atoms with message passing.

Among these, *equivariant GNNs*, highly researched in recent years (Finzi et al., 2020; Fuchs et al., 2020; Huang et al., 2022; Hutchinson et al., 2021b; Satorras et al., 2021b; Liao & Smidt, 2023), explicitly encode the physical symmetries of space: when the input configuration of atoms is translated, rotated, or reflected, the network's scalar and vector outputs transform accordingly (Han et al., 2022). By building in these inductive biases, equivariant models achieve greater data efficiency and generalization in single, static atomic configurations—much as convolutional networks do for images. However, nearly all existing equivariant GNNs ignore the rich temporal context information available in the molecular dynamics simulations data they are often trained on.

In this work we focus on datasets that explicitly expose Molecular Dynamics (MD) trajectories, i.e., time-ordered configurations sampled at a fixed integration time step under a chosen thermodynamic ensemble. This is distinct from geometry relaxations or re-relaxed subsamples (such as revMD17 (Christensen & von Lilienfeld, 2020)), which no longer form a physically meaningful tra-

jectory. While many modern MLIP datasets are constructed from a mixture of protocols (equilibrium databases, rattling, structure searches, etc.), MD-style trajectories remain prevalent and practically important, for example in large-scale benchmarks with MD-like tasks such as OC20/OC22 (Chanussot et al., 2021; Tran et al., 2023) and follow-up challenges. In this paper we investigate: *when MD trajectories are available, how can we best exploit their temporal structure to improve static predictors with minimal information?*

A few recent works have tried to address this by incorporating temporal information, typically by feeding a fixed sequence of consecutive frames into an equivariant spatio-temporal GNN Wu et al. (2023); Satorras et al. (2021a). While these approaches can improve trajectory forecasting, they have two key limitations. First, such models are tied to a fixed history window; they struggle when one wants to predict a single future frame from an arbitrary state, or when the optimal memory length varies across the system. Second, they operate on the assumption that more historical data is always beneficial. This paper challenges that core assumption.

In this work, we propose a different approach. Instead of building a complex spatio–temporal model, we introduce novel training strategy, FRAMES, which utilizes an auxiliary loss function designed to distill temporal information from MD simulations into a standard predictor. This approach is model-agnostic and improves the accuracy of any baseline architecture while leaving it purely static at test time, requiring only a single configuration as input. Furthermore, our framework allows us to systematically investigate the value of temporal information. We challenge the implicit "more is better" assumption, hypothesizing that minimal temporal information—derived from just two consecutive frames—is not only sufficient but optimal. We empirically demonstrate that using more than two frames can be detrimental, degrading model accuracy and efficiency due to data redundancy.

Our contributions are as follows:

- We introduce FRAMES, a novel training strategy using auxiliary loss that leverages temporal data from MD trajectories to significantly improve the accuracy of static energy and force predictors.

- We provide strong empirical evidence for a "less is more" principle, demonstrating that using pairs of two consecutive frames is optimal, while performance degrades with three frames due to data redundancy.

- Our method, applied to a standard Equiformer (Liao & Smidt, 2023) baseline, achieves highly competitive results on the MD17 (Chmiela et al., 2017) and ISO17 (Schütt et al., 2017) benchmarks, validating our approach.

## 2 RELATED WORKS

$SE(3)/E(3)$**-Equivariant Networks.** Incorporating SE(3)/E(3) equivariance (equivariance to 3D rotations, translations, and optionally reflections) as an inductive bias in Graph Neural Networks (GNNs) is often highly beneficial for modeling 3D atomistic systems, leading to strong data efficiency and generalization in many benchmarks. At the same time, recent work has shown that carefully designed non-equivariant or partially equivariant architectures can achieve competitive performance in some regimes, suggesting a spectrum of effective inductive biases rather than a single universally superior choice. (Duval et al., 2023) Key approaches include methods based on irreducible representations (irreps) of the symmetry group, such as Tensor Field Networks (TFNs) (Thomas et al., 2018), SE(3)-Transformers (Fuchs et al., 2020), LieTransformer (Hutchinson et al., 2021a), and Equiformer (Liao & Smidt, 2023) (which FRAMES utilizes). These methods often use spherical harmonics and tensor products to construct equivariant features and operations. Another significant line of work involves scalarization or coordinate-based methods, which operate primarily on invariant quantities (e.g., distances) combined with equivariant directional information. E(n)-Equivariant Graph Neural Networks (EGNNs) (Satorras et al., 2021a) are a prominent example, offering computational efficiency by avoiding higher-order representations. (Garcia Satorras et al., 2021) The Graph Mechanics Network (GMN) (Huang et al., 2022) also employs similar principles for constrained systems. The field strives for a balance between the expressivity of irrep-based models and the efficiency of scalarization techniques, with attention mechanisms also being integrated into equivariant frameworks.

**Equivariant spatio-temporal graph neural networks.** While most GNNs for molecular dynamics assume Markovian dynamics (predicting the next state based only on the current one), real systems exhibit memory effects and periodic motions (Wu et al., 2023). To address this, few equivariant spatio-temporal GNNs have been developed. These models typically process a sequence of past frames to predict future states. For instance, ESTAG (Wu et al., 2023) (Equivariant Spatio-Temporal Attentive Graph Networks) uses historical trajectories and an Equivariant Discrete Fourier Transform (EDFT) to capture non-Markovian properties and periodic patterns. Equivariant Graph Neural Operator (Xu et al., 2024) models dynamics as continuous trajectories using equivariant temporal convolutions. However, such models often rely on a fixed history window at inference, which can be inflexible and computationally demanding. FRAMES differs by using historical frames to improve the training of its latent state via a multi-step lookahead loss, while still allowing for efficient single-step inference without explicit history.

**Multi-step Loss and Auxiliary Predictive Objectives.** Auxiliary tasks, where secondary objectives are learned alongside the primary task, can enhance representation learning and generalization. Predicting future states or properties over multiple steps is a powerful self-supervisory signal that encourages models to capture system dynamics and long-range dependencies. This is a common strategy in reinforcement learning (Merlis et al., 2024) and sequence modeling. In molecular modeling, while some GNNs use multi-step prediction as the primary goal for trajectory forecasting like MDNet (Zheng et al., 2021), or employ other self-supervised tasks like masked position prediction (An et al., 2025), FRAMES specifically uses a multi-step lookahead loss on future energies and forces as an auxiliary objective. The goal is to enrich the GNN's latent representation for improved single-step prediction accuracy and efficiency, rather than direct multi-step forecasting at inference.

Denoising-based objectives are closely related but complementary to our approach. Noisy-node style regularization (Godwin et al., 2022) perturbs equilibrium structures with small random displacements and trains the model to predict the clean configuration, and the recent DeNS method (Liao et al., 2024) applies a similar idea to non-equilibrium structures along its trajectory. These methods operate on unordered or partially ordered sets of structures and do not explicitly exploit full MD trajectories. In contrast, FRAMES leverages the temporal ordering of MD data and shows that, for the benchmarks studied here, minimal temporal context from two consecutive frames already captures most of the useful dynamical signal. We view DeNS and noisy-node-style objectives as complementary: they can be applied in settings without full trajectories and could in principle be combined with FRAMES in future work.

FlashMD(Bigi et al., 2025) proposes direct, long-stride prediction of MD trajectories, taking as input the positions and momenta at a single time step and predicting the configuration at a later time. Their focus is on designing architectures and constraints for fast and stable multi-step MD simulation, whereas our contribution is a training strategy for static predictors that uses an auxiliary temporal loss but leaves inference purely single-frame. Conceptually, their observation that MD is effectively Markovian and can be advanced from the current state alone complements our empirical finding that a very short temporal context (two frames) already provides most of the useful dynamical signal for improving static energy/force prediction.

## 3 METHOD

In §3.1, we formalize the task of energy and force prediction; in §3.2, we describe our model architecture; in §3.3, we detail the FRAMES training objective; and finally, in §3.4, we explain how this framework is used to test our hypothesis on temporal data redundancy.

### 3.1 PROBLEM FORMALIZATION AND PROPOSED APPROACH

The accurate prediction of quantum mechanical properties, such as energy and forces, is essential for modeling complex atomic systems like molecules and crystals. While this paper utilizes datasets generated from Molecular Dynamics (MD) simulations, which consist of atomic trajectories, our primary goal is to enhance predictors that operate on single, static snapshots from these trajectories.

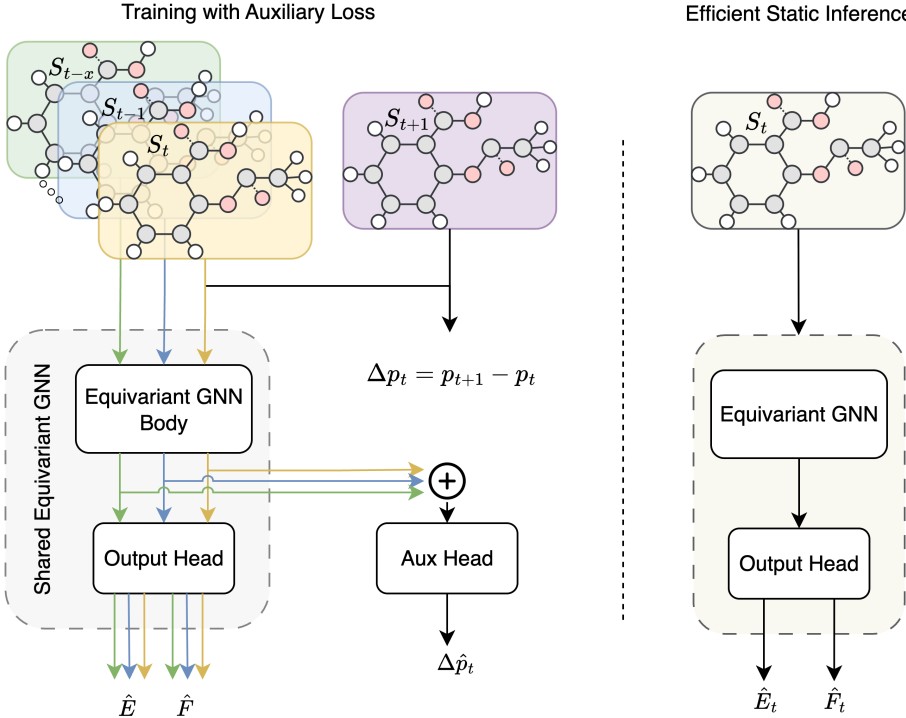

Figure 1: An overview of our proposed training and inference framework. On the left, the shared GNN body processes a history of frames ($S_t$, $S_{t-1}$, etc.) to produce latent embeddings. The primary Output Head uses these embeddings to predict energies and forces for the entire window, supervised by a primary loss. Concurrently, the embeddings are concatenated $\bigoplus$ and fed to an Auxiliary Head, which is trained with an auxiliary loss to predict the displacement to the next frame $\Delta \mathbf{r}_t$. On the right, at test time, the auxiliary head is detached. The model operates as a simple, static predictor, taking a single frame $S_t$ as input to efficiently predict its corresponding energy and forces.

We define an atomic system's configuration at a specific time $t$ as a frame, $S_t$, represented by a set of tuples $S_t = \{(z_i, \mathbf{r}_i) \mid i = 1, \ldots, m\}$ where for each of $m$ atoms, $z_i \in \mathbb{N}$ is its atomic number and $\mathbf{r}_i \in \mathbb{R}^3$ is its 3D position vector.

Associated with each frame $S_t$ is a scalar potential energy shown with $E_t \in \mathbb{R}$, and a set of the atom-wise forces, $F_t = \{\mathbf{f}_i \in \mathbb{R}^3 \mid i \in \{1, \ldots, m\}\}$, where each $f_i$ is the force vector acting on the $i$-th atom.

To avoid confusion with momentum, we use $\mathbf{r}$ for atomic positions throughout.

**Task Definition** We aim to learn the function $f_\phi$, parameterized by $\phi$, which maps a single static frame $S_t$ to its corresponding energy and forces. Formally, the task is:

$$f_\phi(S_t) \to \left\{ \hat{E}_t, \hat{F}_t \right\} \tag{1}$$

where $\hat{E}_t$ and $\hat{F}_t$ are the model's prediction for the true Energy and forces.

**Proposed Approach** Although the system configuration at any given instant determines its energy and, thereby, forces operating on each atom, the temporal evolution of the system overtime provides additional cues over the space of possible energy/force values. However, learning these spatio-temporal dynamics adds to the computational burden, given that the trajectories evolve over long time periods. Instead of relying on complex spatio-temporal models, we aim to capture the useful temporal dynamics from MD trajectories in a lightweight way, so that static predictors can benefit

from temporal information while still operating on single configurations at test time. The key insight is that temporal correlations contain rich cues about energy and forces, but extracting them does not require long histories. In fact, we hypothesize that minimal temporal information — such as pairs of consecutive frames — can be sufficient and likely even more effective than using longer trajectories, which often introduce redundancy and noise. In the following, we describe our model architecture.

## 3.2 MODEL ARCHITECTURE

Our model consists of two main components: a shared GNN Backbone that processes atomic configurations into latent representations, and two distinct Prediction Heads that use these representations to perform the primary and auxiliary tasks (Figure 1).

### 3.2.1 THE GNN BACKBONE

For our GNN backbone, we employ Equiformer architecture (Liao & Smidt, 2023), an $E(3)$-equivariant graph attention transformer. The function of the GNN backbone is to map a single atomic frame, $S_t$, into a set of rich, equivariant latent feature vectors, $h_t$, one for each atom in the system.

During training, this GNN backbone is applied independently to each frame in the input window $(S_{t-T+1}, \ldots, S_t)$ with shared weights, which produces a sequence of embeddings, $(h_{t-T+1}, \ldots, h_t)$ that serves as input to the prediction heads.

### 3.2.2 PREDICTION HEADS

The latent embeddings produced by the GNN backbone are passed to two distinct prediction heads for our multi-task objective.

**Output Head** The Primary Head is responsible for the main prediction task. For each frame $S_t$ in the input window, its corresponding embedding $h_t$ is fed into the Primary Head to produce the predicted energy $\hat{E}_t$ and forces $\hat{F}_t$ for that specific frame. For the scalar value, energy, a feedforward network transforms embedding features $h_t$ on each node into a scalar and then sums over all nodes. The atomic forces $\hat{F}_t$ are then derived analytically as the negative gradient of the predicted energy with respect to the atomic positions, $\hat{F}_t = -\nabla_{\mathbf{r}_t} \hat{E}_t$, ensuring energy conservation.

**Auxiliary Head** Used only during training, the Auxiliary Head's role is to help the model learn from the system's temporal dynamics.

Unlike the Primary Head, the Auxiliary Head takes the concatenated embeddings from the entire historical window of $T$ frames as its single input. This input is the vector $z = [h_{t-T+1}, \ldots, h_t]$. It processes this concatenated vector to predict a single output: the atomic displacement to the next frame, $\Delta \hat{p}_t$.

The Auxiliary Head is itself an equivariant graph attention network, consistent with the GNN backbone. This ensures that the processing of the concatenated temporal information respects the underlying physical symmetries of the system.

## 3.3 THE FRAMES TRAINING OBJECTIVE

To improve the performance of the static predictor defined in §3.1, we introduce a multi-task training objective called FRAMES. This objective is used only during training and combines a standard primary loss with our novel auxiliary loss. The total loss, $\mathcal{L}_{total}$ is a weighted sum of these two components:

$$\mathcal{L}_{total} = \mathcal{L}_{primary} + \lambda_{aux}\mathcal{L}_{aux} \tag{2}$$

where $\lambda_{aux}$ is a hyperparameter that balances the contribution of the auxiliary task. To ensure stable training, all ground-truth energy and force values are normalized before being used in the loss calculations. We now describe each component in detail.

Because the FRAMES objective only augments the loss and does not constrain the backbone architecture, it is directly applicable to a wide range of MLIP models (e.g., Equiformer, NequIP, EGNN). In all experiments below we instantiate FRAMES with Equiformer, but no architectural changes are required to transfer the same objective to other backbones.

### 3.3.1 THE PRIMARY LOSS $\mathcal{L}_{primary}$

The primary loss, $\mathcal{L}_{primary}$ measures the accuracy of the model on the main task of predicting energy and forces. For each of the $T$ frames in the input window, the output head produces the prediction $(\hat{E}_t, \hat{F}_t)$. The primary loss averages error over this entire window, which is a weighted sum of energy and force error:

$$\mathcal{L}_{primary} = \frac{1}{T} \sum_{t'=t-T+1}^{t} \left( \lambda_E |E_{t'} - \hat{E}_{t'}| + \lambda_F \|F_{t'} - \hat{F}_{t'}\|_2 \right) \tag{3}$$

where $\lambda_E$ and $\lambda_F$ are loss-weighting hyperparameters.

### 3.3.2 THE AUXILIARY LOSS $\mathcal{L}_{aux}$

The goal of our auxiliary task is to predict the atomic displacement to the next frame. We define this ground-truth displacement vector, calculated from the simulation data, as:

$$\Delta \mathbf{r}_t = \mathbf{r}_{t+1} - \mathbf{r}_t \tag{4}$$

As described in §3.2, the Auxiliary Head takes the concatenated embeddings, $z = [h_{t-T+1}, \ldots, h_t]$ and outputs a single prediction of this displacement, denoted as $\Delta \hat{p}_t$. Having these two in mind, the auxiliary loss, $\mathcal{L}_{aux}$ is defined as the L2 norm between ground-truth and predicted displacement:

$$\mathcal{L}_{aux} = \|\Delta \hat{p}_t - \Delta \mathbf{r}_t\|_2 \tag{5}$$

By encouraging the model to predict the subsequent motion from the embeddings, this auxiliary task forces the model to learn a representation that is more grounded in the system's physical dynamics, thereby improving performance on the primary task.

Because the FRAMES objective only augments the loss and does not constrain the backbone architecture, it is directly applicable to a wide range of MLIP models (e.g., Equiformer, NequIP, EGNN). In all experiments below we instantiate FRAMES with Equiformer, but no architectural changes are required to transfer the same objective to other backbones.

### 3.4 INVESTIGATING TEMPORAL REDUNDANCY

Our FRAMES framework provides a controlled testbed to investigate the central hypothesis of this work: that for distilling physical priors from dynamics, minimal temporal information is optimal, and that including additional historical data can be detrimental.

To test this hypothesis, we systematically vary the number of historical frames, $T$, used to create the concatenated embedding $z = [h_{t-T+1}, \ldots, h_t]$, for the auxiliary task. We train several otherwise identical models, each with a different value of $T$.

We specifically compare the following three conditions:

- **Baseline** ($T = 1$): This model is trained using only the primary loss, with no auxiliary objective. It represents a standard, purely static predictor.
- **FRAMES** ($T = 2$): Our main proposed model. The auxiliary head is trained on concatenated embeddings from two consecutive frames, providing it with information analogous to velocity.
- **FRAMES** ($T = 3$): A model trained with an auxiliary head fed embeddings from three consecutive frames, providing it with information analogous to acceleration.

Table 1: Mean absolute error results on the MD17 testing set. Energy and force are in units of meV and meV/Å, respectively. This table compares several baseline models against our Equiformer-based approach, which is tested using both two and three frames of temporal context to investigate the effects of data redundancy.

| Model | Aspirin | | Benzene | | Ethanol | | Malonaldehyde | | Naphthalene | | Salicylic acid | | Toluene | | Uracil | |
|---|---|---|---|---|---|---|---|---|---|---|---|---|---|---|---|---|
| | energy | forces | energy | forces | energy | forces | energy | forces | energy | forces | energy | forces | energy | forces | energy | forces |
| SchNet (Schütt et al., 2017) | 16.0 | 58.5 | 3.5 | 13.4 | 3.5 | 16.9 | 5.6 | 28.6 | 6.9 | 25.2 | 8.7 | 36.9 | 5.2 | 24.7 | 6.1 | 24.3 |
| DimeNet (Gasteiger et al., 2020) | 8.8 | 21.6 | 3.4 | 8.1 | 2.8 | 10.0 | 4.5 | 16.6 | 5.3 | 9.3 | 5.8 | 16.2 | 4.4 | 9.4 | 5.0 | 13.1 |
| PaiNN (Schütt et al., 2021) | 6.9 | 14.7 | - | - | 2.7 | 9.7 | 3.9 | 13.8 | 5.0 | 3.3 | 4.9 | 8.5 | 4.1 | 4.1 | 4.5 | 6.0 |
| TorchMD-NET (Thölke & Fabritiis, 2022) | 5.3 | 11.0 | 2.5 | 8.5 | 2.3 | 4.7 | 3.3 | 7.3 | 3.7 | 2.6 | **4.0** | 5.6 | **3.2** | 2.9 | **4.1** | 4.1 |
| NequIP ($L_{max} = 3$) (Batzner et al., 2022) | 5.7 | 8.0 | - | - | **2.2** | **3.1** | 3.3 | 5.6 | 4.9 | 1.7 | 4.6 | 3.9 | 4.0 | 2.0 | 4.5 | **3.3** |
| Equiformer | 5.3 | 7.2 | **2.2** | 6.6 | **2.2** | **3.1** | 3.3 | 5.8 | 3.7 | 2.1 | 4.5 | 4.1 | 3.8 | 2.1 | 4.3 | 3.8 |
| Equiformer+Noisy Nodes | 10.5 | 8 | 4.3 | 6.3 | 2.6 | 3.8 | 3.6 | 6.3 | 3.6 | 2.5 | 6 | 5.2 | 4.3 | 2.3 | 6.5 | 5.5 |
| Equiformer + 2 Frames | **5.2** $_{\pm 0.16}$ | **7.0** $_{\pm 0.09}$ | **2.4** $_{\pm 0.07}$ | 6.3 $_{\pm 0.27}$ | **2.2** $_{\pm 0.02}$ | 3.2 $_{\pm 0.05}$ | **3.3** $_{\pm 0.05}$ | **5.6** $_{\pm 0.17}$ | **3.6** $_{\pm 0.03}$ | 2.2 | $\underline{4.3}$ $_{\pm 0.25}$ | 4.1 $_{\pm 0.04}$ | $\underline{3.6}$ $_{\pm 0.03}$ | **2** $_{\pm 0.9}$ | **4.1** $_{\pm 0.12}$ | 3.5 $_{\pm 0.10}$ |
| Equiformer + 3 Frames | 5.3 | 7.3 | 2.6 | **6.1** | 2.2 | 3.5 | 3.3 | 6 | 3.8 | 2.4 | 4.4 | 4.4 | $\underline{3.5}$ | 2 | 4.1 | 3.9 |

Crucially, while the models are trained differently, they are all evaluated on the exact same task at inference time: the accuracy of static energy and force prediction on the test set, using only a single frame $S_t$ as input. The performance on this final task will be used to validate our hypothesis.

To ensure a fair and controlled comparison, all other aspects of the experimental setup are held constant across these cases. This includes the core model architecture, the training objective (the combined primary and auxiliary loss function), and all hyperparameters.

## 4 EXPERIMENTS

To validate our proposed FRAMES framework and test our hypothesis on temporal data redundancy, we conduct a series of experiments on standard benchmarks. We begin in §4.1 by evaluating our primary results on the widely-used MD17 dataset, comparing our method against several state-of-the-art baselines. Then in §4.1 we present a key ablation study to justify our choice of auxiliary objective. Finally, we test the generalization of our findings on the ISO17 dataset in §4.2 and provide an illustrative example on a spring-mass system in §4.3 which helps provide insights into the phenomenon.

### 4.1 MD17 DATASET

**Dataset.** The MD17 dataset (Chmiela et al., 2017) features ab-initio molecular dynamics trajectories for 8 small organic molecules, including Aspirin and Toluene. The primary task is to predict the potential energy and inter-atomic forces for each molecular configuration (frame) in a trajectory. Following standard benchmarks, we use 950 frames for training and 50 for validation, with the remainder used for testing. Crucially for our temporal analysis, we ensure that training samples are drawn sequentially with a fixed time lag of $\Delta_t$ between them, preserving the physical dynamics of the original simulation.

**Implemetation details.** We train Equiformer (Liao & Smidt, 2023) with FRAMES based on the official implementation. We trained this model once with two frames as input, and once with three frames as input. We also implemented a noisy-node style auxiliary loss, where the model predicts atomic displacements from small random perturbations of the current structure, and considered it as another usefull baseline. Further details of the noisy-node baseline are provided in Appendix A.5.

**Main Results.** The results, presented in Table 1, strongly support our central hypothesis. The *Equiformer + 2 Frames* model, which is supplied with velocity information, consistently outperforms the standard *Equiformer* $(T = 1)$ baseline across nearly all molecules, achieving the best force prediction on 5 out of 8 molecules. In contrast, the *Equiformer + 3 Frames* model, which implicitly includes acceleration data, shows a marked degradation in performance. For instance, in molecules like Benzene and Malonaldehyde, its performance on force prediction is worse than the $T = 2$ model and is comparable or worse than the T=1 baseline. This trend suggests that adding further temporal context beyond velocity introduces redundant information, which, akin to multicollinearity, hinders the model's ability to learn the underlying force field effectively.

**Ablation Study.** We conducted an ablation study to empirically validate our choice of the auxiliary learning objective, as defined in §3.3. We compare two distinct auxiliary loss formulations for

our FRAMES (T=2) model. The first is our proposed method, which uses a loss on the predicted displacement, $\mathcal{L}_{aux} = \|\Delta \hat{p}_t - \Delta \mathbf{r}_t\|_2$. The second is a more conventional alternative, which uses a loss on the predicted energy and forces of the next frame, $\mathcal{L}'_{aux} = \lambda_E |E_{t+1} - \hat{E}_{t+1}| + \lambda_F \|F_{t+1} - \hat{F}_{t+1}\|_2$.

The results, presented in Table 2, show that both objectives provide a significant improvement over the baseline, with highly competitive overall performance. While predicting future forces and energies yields marginally better results on some molecules (e.g., Benzene), our proposed displacement prediction objective achieves superior or equivalent performance on the majority of the benchmark, including on larger molecules like Aspirin and Salicylic Acid.

Given that displacement prediction offers more consistent performance across the benchmark and represents a more direct and fundamental dynamic property (the immediate consequence of the current frame's forces), we confirm its effectiveness and select it as the default auxiliary objective for our FRAMES framework.

Table 2: Ablation study on the choice of auxiliary loss for the $T = 2$ model. We compare our proposed method, which uses a loss on atomic displacements (Aux: $E_{t+1}, F_{t+1}$) against our proposed method of predicting atomic displacements (Aux: $\Delta \mathbf{r}_t$). Results are mean absolute error (MAE) in meV for energy and meV/Å for forces.

| Model (T=2) | Aspirin | | Benzene | | Ethanol | | Malonaldehyde | | Naphthalene | | Salicylic acid | | Toluene | | Uracil | |
|---|---|---|---|---|---|---|---|---|---|---|---|---|---|---|---|---|
| | energy | forces | energy | forces | energy | forces | energy | forces | energy | forces | energy | forces | energy | forces | energy | forces |
| Equiformer (T=1, Baseline) | 5.3 | 7.2 | 2.2 | 6.6 | 2.2 | 3.1 | 3.3 | 5.8 | 3.7 | 2.1 | 4.5 | 4.1 | 3.8 | 2.1 | 4.3 | 3.8 |
| FRAMES (T=2) with Aux: $E_{t+1}, F_{t+1}$ | 5.3 | 7.2 | **2.3** | **6.1** | **2.2** | 3.2 | **3.2** | **5.6** | **3.6** | 2.2 | 4.4 | 4.2 | 3.8 | **1.9** | **4.1** | 3.6 |
| FRAMES (T=2) with Aux: $\Delta \mathbf{r}_t$ | **5.2** | **7.0** | 2.4 | 6.3 | **2.2** | 3.2 | 3.3 | **5.6** | **3.6** | 2.2 | **4.3** | **4.1** | **3.6** | 2.0 | 4.2 | **3.5** |

## 4.2 ISO17 DATASET

**Dataset.** We further validate our approach on the ISO17 dataset (Schütt et al., 2017). This dataset contains molecular dynamics trajectories of 129 isomers of $C_7O_2H_{10}$, presenting a different challenge by testing generalization across constitutional isomers. As described in the original work, the dataset is split into two evaluation scenarios. The first, which we term "Within Distribution," tests for generalization to unseen conformations of molecules that were included in the training set. The second, more challenging "Outside Distribution" scenario tests for generalization to entirely new molecular structures (isomers) that the model has never seen during training.

**Results.** The results, presented in Table 3, demonstrate the remarkable generalization capability of our FRAMES framework. Our FRAMES (T=2) model achieves the best performance by a significant margin across all four evaluation metrics. On the "Within Distribution" task, it substantially improves upon the baseline, confirming that our method learns a more accurate potential energy surface. More importantly, on the challenging "Outside Distribution" task, FRAMES (T=2) shows a dramatic improvement in generalizing to unseen isomers, indicating that the physical priors learned via the auxiliary loss are not molecule-specific. Consistent with our findings on MD17, the FRAMES (T=3) model shows a clear degradation in performance, often performing worse than the baseline. This validates our central hypothesis that minimal temporal information is optimal and that data redundancy hinders generalization, even across different chemical structures.

## 4.3 SPRING-MASS

To build intuition for our hypothesis, we analyze a simple spring–mass system where the underlying physics is known. This toy problem allows us to create a controlled environment to illustrate the effects of data redundancy when predicting forces (equivalently, accelerations since $m=1$) from trajectory data, a scenario directly analogous to the multicollinearity problem in linear regression. We test both a simple linear regressor and a non–linear predictor on this problem to demonstrate how redundant temporal information affects both direct estimation and more complex representation learning.

**Implementation.** We simulate a simple harmonic oscillator governed by Hooke's Law, $F = -kx$, setting mass $m = 1.0$ and spring constant $k = 1.0$. The trajectory is generated by numerically

Table 3: Mean Absolute Error on the ISO17 test sets. Our FRAMES (T=2) model is compared against the baseline Equiformer and a T=3 model. Within Distribution tests generalization to new conformations of known molecules, while Outside Distribution tests generalization to entirely new isomers.

|  | Within Distribution | | Outside Distribution | |
|---|---|---|---|---|
|  | Energy | Forces | Energy | Forces |
| Equiformer (Baseline) | 0.13228 | 0.0093 | 0.1460 | 0.0174 |
| FRAMES ($T = 2$) | **0.00569** | **0.0053** | **0.0248** | **0.0154** |
| FRAMES ($T = 3$) | 0.07009 | 0.0101 | 0.0639 | 0.0187 |

integrating the equation of motion $\ddot{x} = -x$ to produce a time series of positions $\{\mathbf{r}_t\}$. Since $m=1$, the force and acceleration coincide, so predicting $F_t$ is equivalent to predicting the acceleration.

For this synthetic dataset, we consider two simplified models: a linear model and a non–linear MLP. We randomly sample from the trajectories generated according to the above setting, and train on 8000 samples for the non–linear model and 100 samples for the linear model, with 2000 samples reserved for testing.

The *linear model* consists of a single shared linear layer followed by two linear heads: a main head that predicts the target force $F_t$ and an auxiliary head that predicts the FRAMES objective, the next–step displacement $\Delta\mathbf{r}_i$. The *non–linear model* has the same structure (shared body, main head, auxiliary head), but each component is implemented as a small Multi–Layer Perceptron (MLP) instead of a single linear layer.

For $T = 1$, the baseline, we disable the auxiliary head and train both models only with the primary loss on the current force $F_t$, which corresponds to a standard static predictor without FRAMES. For $T > 1$, both models are trained using the FRAMES objective. The input for a history of $T$ is the vector of positions $[\mathbf{r}_{t-T+1}, \ldots, \mathbf{r}_t]$. The primary task for both models is to predict the force $F_t$ at the current time. For the auxiliary task, a simple linear head takes the concatenated hidden layer representations (embeddings) from the historical window and is trained to predict the next-step displacement, $\Delta\mathbf{r}_t = \mathbf{r}_{t+1} - \mathbf{r}_t$.

**Results.** The results, for the nonlinear model is summarized in Table 4 and for the linear model visualized in Figure 2, which both of them clearly support our central hypothesis. Performance is extremely poor with one frame ($T = 1$), as a single position does not contain information on temporal dynamics. The error decreases significantly for the ($T = 2$) model, which can infer the velocity, but increases again for ($T = 3$). This suggests that while minimal temporal information is highly beneficial, additional frames introduce redundancy that degrades performance. This simple example confirms the core principle that "less is more," that we also observe in our main experiments on complex molecular systems.

Table 4: Mean Squared Error (MSE) on the spring-mass toy example. $T$ denotes the number of historical frames used as input to predict the current force.

| Model | T=1 | T=2 | T=3 | T=4 | T=5 | T=6 |
|---|---|---|---|---|---|---|
| MLP Model ($10^{-9}$) | $1.24_{\pm 1.02}$ | $0.83_{\pm 0.44}$ | $1.55_{\pm 1.09}$ | $1.28_{\pm 0.72}$ | $2.15_{\pm 1.38}$ | $1.05_{\pm 0.45}$ |

## 5 CONCLUSION

In this work, we addressed the challenge of improving molecular force and energy prediction by leveraging temporal information from Molecular Dynamics simulations. We introduced FRAMES, a novel and model-agnostic auxiliary loss that distills physical priors from pairwise frame dynamics into a predictor that remains purely static and efficient at inference time. Our experiments on the MD17 and ISO17 benchmarks demonstrate that FRAMES significantly improves the accuracy of a strong Equiformer baseline, achieving highly competitive results in both energy and force prediction.

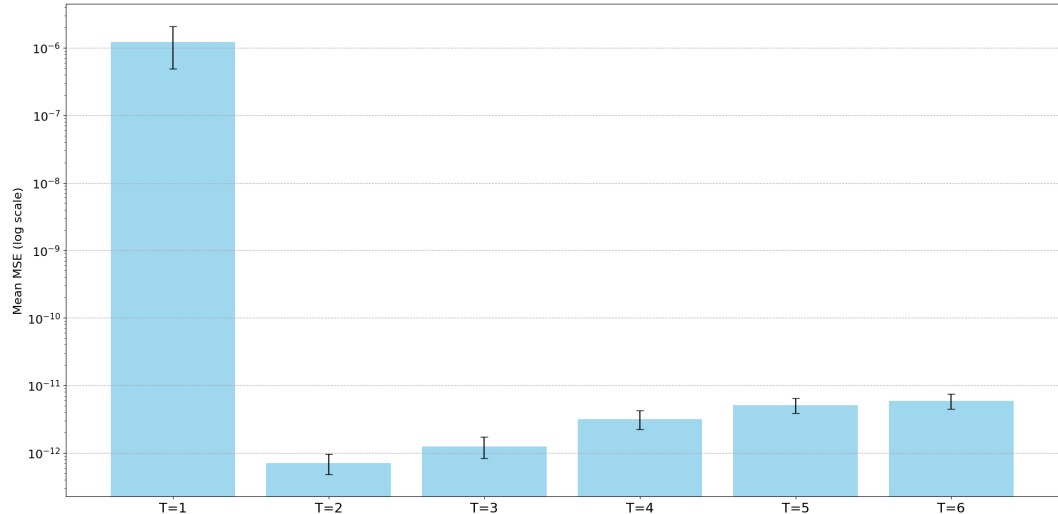

Figure 2: Mean squared error (MSE) of the linear predictor on the spring–mass toy system as a function of the history length $T$ (x-axis). The plot highlights the large performance improvement when moving from $T=1$ to $T=2$, followed by degradation once redundant temporal information ($T \geq 3$) is included. A 95% confidence interval is shown.

Furthermore, we provided strong empirical evidence for our "less is more" hypothesis. We showed that using minimal temporal information from two consecutive frames is optimal for this task, while including more historical data in the training procedure can be detrimental, degrading model performance due to data redundancy. This finding was validated across complex molecular benchmarks and an intuitive spring-mass toy example.

Future work could explore the application of the FRAMES objective to a wider range of equivariant architectures and other scientific domains where simulation trajectories are available. Ultimately, our work highlights a simple, powerful, and computationally efficient strategy for creating more physically-grounded and accurate molecular predictors.

## ETHICS STATEMENT

This work proposes a training strategy for graph neural networks aimed at improving energy and force prediction in atomic and molecular systems. The research is entirely computational and does not involve human subjects, personal data, or sensitive information. The datasets used (e.g., MD17, ISO17) are publicly available and widely adopted benchmarks in the community. We believe that the outcomes of this work will have positive impacts by advancing the use of AI for scientific discovery, particularly in molecular modeling and materials design. We do not foresee significant risks of misuse or negative societal impacts beyond those already inherent to general machine learning research in molecular simulations.

## REPRODUCIBILITY STATEMENT

We have taken several steps to ensure the reproducibility of our work. Details of the FRAMES datasets and implementation details are described in the main text in Section 4, with further training hyperparameters details provided in the Appendix 5. We include results across multiple random seeds to demonstrate stability, and ablation studies to clarify the contribution of individual components. To facilitate replication, we have released anonymized source code and scripts for training and evaluation here https://anonymous.4open.science/r/FRAMES-7AB9/README.md.

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

# A APPENDIX: EXPERIMENTAL DETAILS

Our implementation is based on the official open-source code for Equiformer (Liao & Smidt, 2023). For hyperparameters shared with the original work, we adopt their reported values unless otherwise specified to ensure a fair comparison. All models were trained using an Adam optimizer with an initial learning rate of $5 \times 10^{-4}$.

## A.1 TRAINING PROCEDURE

For the FRAMES models, the auxiliary loss weight, $\lambda_{aux}$, was linearly decayed from its initial value (see Table 5) to 0 over the course of training. To manage the memory requirements of processing historical data, we adjusted the batch sizes. The baseline model ('T=1') used a batch size of 8. For **FRAMES**, we used a batch size of 4 for both the 'T=2' and 'T=3' configurations.

## A.2 TRAJECTORY SUBSAMPLING FOR MD EXPERIMENTS

For all MD-based experiments (MD17 and ISO17), we do not feed every raw MD frame directly to the model. Instead, for each molecule we construct shorter sub-trajectories by uniformly subsampling frames along the original trajectory. Let $(S_1, \ldots, S_L)$ denote the sequence of configurations for a given molecule. We choose a stride $k \geq 1$ and build training windows of length $T$ as

$$(S_t, S_{t+k}, S_{t+2k}, \ldots, S_{t+(T-1)k}),$$

so that consecutive frames inside a window are equally spaced and separated by $k$ steps in the original MD trajectory. The stride $k$ is chosen automatically for each trajectory based on its length and the desired number of training samples, so that we obtain approximately the target number of windows while keeping the frames in each window well separated (typically on the order of tens of MD steps).

Note that although the frames in a sub-trajectory are non-adjacent in the original MD sequence when $k > 1$, the FRAMES auxiliary target is always defined on adjacent elements of the *subsampled* window. Concretely, if $(S_t, S_{t+k}, \ldots)$ is a window, we predict the displacement $\Delta \mathbf{r}_j = \mathbf{r}_{j+1} - \mathbf{r}_j$ between consecutive frames inside this subsampled sequence, where $\mathbf{r}_j$ and $\mathbf{r}_{j+1}$ correspond to configurations that are $k$ integration steps apart in the underlying MD trajectory.

## A.3 MD17 HYPERPARAMETERS

The key loss coefficients for our experiments on the MD17 dataset are detailed in Table 5.

Table 5: Hyperparameters used for training our 'FRAMES' models on the MD17 dataset. We report the coefficients for the primary loss ($\lambda_E, \lambda_F$) and the initial value for the auxiliary loss ($\lambda_{aux}$).

| Hyper-parameter | Aspirin | Benzene | Ethanol | Malonaldehyde | Naphthalene | Salicylic acid | Toluene | Uracil |
|---|---|---|---|---|---|---|---|---|
| Energy coefficient $\lambda_E$ | 1 | 1 | 1 | 1 | 2 | 1 | 1 | 1 |
| Force coefficient $\lambda_F$ | 80 | 80 | 80 | 100 | 20 | 80 | 80 | 20 |
| FRAMES coefficient $\lambda_{aux}$ | 1 | 0.25 | 0.25 | 0.25 | 1 | 1 | 1 | 0.25 |

## A.4 ISO17 HYPERPARAMETERS

For the ISO17 experiments, we used a consistent set of hyperparameters across all isomers: the energy coefficient $\lambda_E = 1$, the force coefficient $\lambda_F = 80$, and the initial auxiliary loss coefficient $\lambda_{aux} = 0.25$. The model architecture and training procedure were kept identical to those used for the MD17 experiments.

## A.5 NOISY-NODE AUXILIARY OBJECTIVE

For the noisy-node baseline, we follow the general idea of Godwin et al. (2022) and add an additional denoising-style auxiliary loss on top of the main energy/force prediction loss. Concretely, during training we apply the following procedure independently for each graph (configuration): with probability $p_{\text{noise}} = 0.1$ we construct a corrupted version of the input by randomly selecting a

fraction $p_{\mathrm{corr}} = 0.25$ of the atoms and perturbing their positions with small Gaussian noise of standard deviation $0.02$ (in the same units as the input coordinates). The backbone GNN encodes this corrupted structure, and an auxiliary head is trained to predict the displacement between the clean and noisy positions (i.e., to denoise the perturbed atoms). The auxiliary noisy-node loss is combined with the main loss using a fixed weight $\lambda_{\mathrm{aux}} = 5$, so that the total objective is

$$L_{\mathrm{total}} = L_{\mathrm{main}} + \lambda_{\mathrm{aux}}\, L_{\mathrm{noisy\text{-}node}}.$$

This auxiliary head is used only during training; at inference time the model reduces to the standard single-frame predictor without any denoising branch.

