# OpenReview forum: "Less is More: Improving Molecular Force Fields with Minimal Temporal Information"
_ICLR.cc/2026/Conference — Submitted to ICLR 2026_

### Official Review · Reviewer_ZeyN · 2025-10-26

**Soundness:** 3
**Presentation:** 3
**Contribution:** 2
**Rating:** 4
**Confidence:** 5

**Summary:**

The authors propose FRAMES, a training strategy for machine-learned interatomic potentials (MLIPs) where trajectory data is learned at the same time as energies and forces, thereby increasing the amount of geometrical and physical information available to the model and improving prediction accuracies for the MLIP. The method is employed to improve performance on the MD17 and ISO17 datasets.

**Strengths:**

Perhaps needless to say, both MLIPs and trajectory predictor models are known in the literature. Using them together to improve MLIP accuracies is however an original and interesting idea. The quality of the investigation is high, all design choices and experiments are reasonable and justified, and the clarity of the presentation is excellent.

**Weaknesses:**

In my opinion, the work has a single major weakness, which unfortunately prevents it from having a wide significance in the field. Modern datasets for MLIPs are not generally obtained from molecular dynamics simulations. While this dataset construction technique was popular around 2017 (although, even at that time, it was never the one primarily used by actual practitioners in chemistry), it is now at best used as one of the many splits that go into a well-crafted dataset. More often than not, it is not used at all. Popular alternatives include sampling from equilibrium databases, rattling of equilibrium structures, cutting (to obtain surfaces), random structure searches, random defect creation, random element substitutions, cell distortions and so on. Maybe for this reason, the authors limit their investigation to two relatively old datasets (MD17, ISO17) which would definitely not be used to train state-of-the-art models today. As such, I believe the evaluation is lacking, but I do not know if improving it is possible, given that recent datasets cannot easily be trained with the proposed strategy. A similar recent idea, denoising non-equilibrium structures (DeNS, which is incidentally not referenced by the authors), is struggling to be used consistently despite its applicability to arbitrary (including modern) datasets. This is due to its modest gains in accuracy (which are comparable to those shown here), compared to a moderately tedious implementation (also comparable to the architecture developed in this work). As such, I do not realistically see many people in the field benefitting from FRAMES.

Other weaknesses:
- In the abstract, the sentence "MD generates trajectories of atomic positions of molecular systems moving from higher energy states to lower energy stable/equilibrium states" is incorrect. This is not geometry optimization. If this were true, MD would eventually have to lose all its energy and freeze. In short, in molecular dynamics, it is not the positions that are updated with something proportional to the forces, but the momenta. This creates a much more complex dynamical system.
- The literature review section is lacking in many aspects. Recent evidence points to the fact of "incorporating E(3)-equivariance" not being "crucial" (line 84) for MLIPs. On the contrary, unconstrained models can achieve similar accuracies and at least equal computational efficiency compared to equivariant models. In the same section, something called "TEMPO" (line 87) is mentioned. I can only suppose this was an earlier name for FRAMES. As mentioned earlier, DeNS deserves a mention as a related piece of work, trying to achieve a similar effect to the current work. FlashMD, a modern variant of MDNet, provides a good analysis to what the authors correctly identify as the need for only two structures to predict the positions of the new frame. Finally, recent literature on architectures for MLIPs in chemistry and materials is omitted, although I can totally understand if this is due to the page limit.
- Comparison to state-of-the-art architectures other than Equiformer is very limited, and, once again, this is probably due to the fact that only MD-derived datasets are usable for this method. Nonetheless, the authors could try their method to improve a different architecture for completeness.

**Questions:**

I have no questions for the authors at this stage. If they see fit, they can elaborate on the points above and we can discuss them more in detail.

---

> ### Author Response · Authors · 2025-11-29
>
> We thank the reviewer for the thoughtful feedback. We address the main concern about the scope and relevance of MD-based datasets first, then respond point-by-point to the specific comments.
>
> Main concern: reliance on MD datasets and practical relevance
> > **Modern datasets for MLIPs are not generally obtained from MD simulations...**
>
> We agree that the broader MLIP community now uses a variety of data-generation protocols beyond straightforward MD trajectories: equilibrium databases, rattling, random structure searches, defect creation, cell distortions, etc. We would like to clarify that:
>
> - MD-style trajectories are still prevalent and practically important.
> Large-scale benchmarks such as **OC20/OC22** and follow-up challenges (e.g., **ODAC23**) explicitly include MD-like tasks and provide access to trajectories (e.g., IS2RS, MD relaxations) even when the training splits are shuffled. There is an active line of work on improving models for exactly these tasks, including competition tracks and follow-up papers. FRAMES targets precisely this setting: when MD trajectories are available, how can we best leverage their temporal structure to improve static predictors?
>
> > **A similar recent idea, denoising non-equilibrium structures is struggling...**
>
> We appreciate the pointer to DeNS and agree it deserves explicit discussion. We will add DeNS in the related work and clearly position similarities and differences.
>
> Our goal in this work is not to claim that FRAMES dominates DeNS, but to show that given MD trajectories, carefully leveraging just two consecutive frames can be surprisingly more effective than using longer histories. Our main contribution lies in utilizing the trajectory information via a simple loss while also deriving the insight that full tragjectory introduces data redundancy.
>
> Implementing a full DeNS baseline within our Equiformer codebase would require non-trivial engineering, and a fair, apples-to-apples comparison across architectures and datasets is beyond the scope of this submission. Instead, we will:
>
> - add a dedicated paragraph in Related Work summarizing DeNS and noisy nodes, and explicitly highlight DeNS as a complementary direction that operates on a broader class of datasets (and thus could potentially be combined with FRAMES in future work).
>
> We hope this clarifies why we see FRAMES as complementary to, rather than competing directly with, DeNS.
>
> >  - **In the abstract, the sentence "MD generates trajectories ...**
>
> Thank you for the suggestion. We have updated the manuscript accordingly.
>
> > - **The literature review section is lacking in many aspects. Recent evidence points to the fact of "incorporating E(3)-equivariance" not being "crucial" (line 84) for MLIPs...**
>
> We agree that “crucial” is too strong. Our intention was to emphasize that equivariant models have shown strong data-efficiency and accuracy on many benchmarks. We will soften the wording, and briefly mention that recent non-equivariant or less-constrained models can achieve comparable performance in some regimes, with explicit references added.
>
> > - **FlashMD, a modern variant of MDNet, provides a good analysis...**
>
> Thank you for this pointer — we were not aware of FlashMD. We will cite it and highlight in Related Work how their analysis about two-structure inputs complements our empirical results that two frames are optimal in our auxiliary-loss setting. Conceptually, their work focuses on efficient forecasting architectures, while our contribution is a training strategy that leaves inference static and explores temporal redundancy.
>
> **We also note and appreciate that the reviewer agrees with the core idea  that only two structures are likely needed to predict the positions of the new frame.** This significant insight of our work sheds some light into the underlying phenomenon which can improve efficiency of the modelling.
>
> > - **Recent literature on architectures for MLIPs in chemistry and materials is omitted...**
> > - **Comparison to state-of-the-art architectures other than Equiformer is very limited...**
> > - **The authors could try their method to improve a different architecture for completeness.**
>
> We agree that our related-work section is compressed due to space constraints. We have updated the manuscript to:
>
> - extend the discussion of recent MLIP architectures (beyond the ones already in Table 1), and make clearer in the text that Table 1 already includes comparisons to strong baselines such as SchNet, DimeNet, TorchMD-Net, and NequIP, where our Equiformer+FRAMES model is competitive and often achieves the best force MAEs across molecules.
>
> - Regarding “another architecture”: empirically, FRAMES is architecture-agnostic as it acts via an auxiliary loss. Due to computational constraints we focused on Equiformer, but the method can be plugged into other backbones (e.g., NequIP or EGNN) without structural changes; we emphasize this more clearly and discuss multi-architecture validation as future work.

---

### Official Review · Reviewer_8Ljb · 2025-10-28

**Soundness:** 1
**Presentation:** 2
**Contribution:** 2
**Rating:** 2
**Confidence:** 4

**Summary:**

The authors suggest adding an auxiliary prediction head to MLIPs during training only, that takes in embeddings from multiple geometries and predicts the atomic displacement $\Delta p = p_{t+1} - p_t$.
They test the idea using EquiformerV1 on ISO17, MD17, and a toy harmonic oscillator.

**Strengths:**

- The idea is simple to implement and cheap
- The suggested auxiliary loss improves forces, and especially the energies, on ISO17

**Weaknesses:**

- Weak results on MD17 (table 1). The loss only improved the accuracy over the baseline in 9/16 cases, each improvement being less than 5%.
- I am not sure the small gains have anything to do with temporality. The auxiliary loss seems like a reconstruction loss.

**Questions:**

- Can you add ablations with (a) predicting the atomic displacement to a small random perturbation instead of $p_{t+1}$ (predicting $\Delta p = p_t^{noised} - p_t$) (b) just reconstructing the current input geometry p_t (only taking in a single geometry, predicting $p = p_t$)? These "noisy node" auxillary losses are known in the literature to help model performance and are easier to implement. Thus the proposed new method should beat this baseline
- To improve the generality of the empirical results, can you show results for direct-force prediction, e.g. with the otherwise similar EquiformerV2?
- I think the synthetic toy benchmark of the harmonic oscillator is missing the MSE of predicting the force without using the auxiliary loss as a baseline
- Figure 2 needs to be revised with axis labels, clearer x-tick labels, and larger fonts
- “MD generates trajectories of atomic positions of molecular systems moving from higher energy states to lower energy stable/equilibrium states.” I think you are referring to “relaxations”. The usual microcanonical NVT-ensemble MD conserves total energy.

---

> ### Author Response · Authors · 2025-11-29
>
> We thank the reviewer for the careful assessment and concrete experimental suggestions. We address the weaknesses and questions in turn.
>
> > **Weak results on MD17 (table 1). The loss only improved the accuracy over the baseline in 9/16 cases, each improvement being less than 5%.**
>
> We respectfully disagree. MD17 is a heavily studied and saturated benchmark, and our Equiformer baseline is very strong. In this regime, average improvements of a few percent in force MAE can still support promising ideas and provide significant insights into the phenomenon. We also emphasize that:
>
> - FRAMES comes with no additional inference cost: at test time the model is identical to the single-frame Equiformer and uses one configuration.
> - The gains are more pronounced in other settings: on ISO17, FRAMES (T=2) reduces energy MAE by more than a factor of 5 in the within-distribution and ~6× in the out-of-distribution split, compared to the same Equiformer baseline. So the method is not limited to marginal improvements.
>
> > **I am not sure the small gains have anything to do with temporality. The auxiliary loss seems like a reconstruction loss.**
>
> Again, we respectfully disagree. Our auxiliary task is not a simple reconstruction of the current frame. It predicts the displacement vector between frames, $\Delta p_t = p_{t+1} - p_t$, based on a history window of $T$ frames, i.e., the delta between the current and next frame. In reconstruction losses, the prediction is on the same frame; in contrast, our use of multiple frames for both input and prediction clearly **incorporates temporal information**. For $T=2$, this task is closely tied to velocity and therefore directly encodes temporal information. This is further supported by our synthetic spring–mass experiment, which studies temporal information in a simple setting.
>
> > 1. **Can you add ablations with (a) predicting the atomic...**
>
> We appreciate your suggestions. In the revision, we will report:
>
> (b) Single-frame baseline:
> This is exactly the T=1 baseline already included: the model receives only a single frame, predicts its energy/forces, and no temporal auxiliary loss is used.
>
> (a) Noisy-node / random-perturbation auxiliary loss:
>
> Following your suggestion, we implemented a “noisy nodes” style auxiliary objective where the model predicts atomic displacements from small random perturbations of the current structure. As in Godwin et al. [1], this regularization provides limited or no benefit on MD17 and is mainly helpful on equilibrium datasets. In contrast, our FRAMES auxiliary loss, which uses true temporal displacements between consecutive MD frames, yields systematic improvements over the same Equiformer baseline on MD17 and ISO17 and is the only variant that consistently exhibits the “2 frames good, 3 frames bad” behaviour. This supports our claim that minimal temporal context—rather than generic denoising—is the key ingredient.
>
> We will include the new baseline in the revision with a short discussion highlighting that FRAMES is competitive with or better than the noisy-node variant. This baseline currently yields:
>
> |Model|Toluene||Uracil||
> |-|-|-|-|-|
> ||Energy|Forces|Energy|Forces|
> |Equiformer + Noisy Nodes|4.3|2.3|5.5|6.5|
> |Equiformer + Frames|3.6|2.0|4.1|3.5|
>
> The units are in $meV$.
>
> > 2. **To improve the generality of the empirical results,... with the otherwise similar EquiformerV2?**
>
> Our current experiments are based on the original Equiformer (V1) implementation. We follow the standard MD17/ISO17 setup, where the model is trained and evaluated on forces, so FRAMES is already validated in a direct-force prediction setting, albeit on EquiformerV1 rather than V2.
>
> The FRAMES objective itself is architecture-agnostic: it only adds an auxiliary temporal loss on top of an existing force-prediction backbone and can, in principle, be applied to EquiformerV2 (or other MLIP architectures) without modification. Within rebuttal-time and compute limits, we plan to report a small set of additional experiments with EquiformerV2 in the appendix (if they complete in time); a more exhaustive EquiformerV2 study is left for future work.
>
> > 3. **Toy benchmark of the harmonic oscillator is missing the MSE of predicting the force ...**
> > 4. **Figure 2 needs to be revised...**
>
> We thank the reviewer for your suggestions.
>
> 3.  This is exactly the T=1 (no auxiliary loss) baseline already included in the spring-mass experiment.
> 4. In the revision, we will improve Figure 2. Revise Figure 2 with clear axis labels, more readable x-ticks, and larger fonts.
>
> > 5. **"MD generates ... states." I think you are referring to “relaxations”. ...**
>
> We agree. We have corrected this text in the updated manuscript to refer explicitly to MD under standard ensembles (e.g., NVT) and distinguish it from geometry relaxations / energy minimization.
>
> ### References:
> [1] Jonathan Godwin, et al. "Simple GNN Regularisation for 3D Molecular Property Prediction & Beyond" arXiv preprint arXiv:2106.07971 (2021).

---

### Official Review · Reviewer_acmh · 2025-10-31

**Soundness:** 2
**Presentation:** 2
**Contribution:** 3
**Rating:** 6
**Confidence:** 4

**Summary:**

This paper proposes FRAMES, a new training strategy for machine-learning interatomic potentials (MLIPs) on molecular dynamics (MD) trajectories: In addition to the usual approach of predicting energies and forces for each frame independently, an additional head is tasked with predicting the positions at the next step, improving the quality of internal representations and therefore accuracy. The authors further investigate how much temporal context is beneficial and observe that using two frames improves accuracy, while three frames degrade performance. Results are presented on MD17, ISO17, and a toy spring–mass system.

Note: An LLM (ChatGPT 5) was used to expand the review from notes to the full text. The model did not suggest useful feedback, and therefore did not contribute beyond this task.

**Strengths:**

- A simple and appealing idea, clearly explained and easy to implement, with practical relevance for MLIP training.
- The method is model-agnostic, requires no additional inference-time cost, and appears practically useful.
- Experimental results generally support the claim that using minimal temporal information can improve performance.
- Error bars are included for some results, which is appreciated.
- The spring–mass example illustrates the intended intuition.

**Weaknesses:**

- Statistical signal is weak in several experiments and some cases with error bars show high variance. This makes the "less is more" conclusion, and the overall claim of FRAMES providing a decisive advantage, feel somewhat premature.
- The use of original MD17, rather than revMD17 (https://archive.materialscloud.org/records/pfffs-fff86; DOI:10.1088/2632-2153/abba6f) introduces known noise issues; the conclusions may be confounded by simulation artefacts rather than true learning dynamics.
- The paper occasionally overstates its claims. Observing that three frames perform worse than two does not necessarily imply that more temporal information is inherently harmful in general.
- The method is limited to deterministic, fixed–time step MD trajectories (NVE). It would not apply to data from stochastic thermostats or irregular sampling. In practice, due to their inherent correlations, data from MD is not often used directly for MLIP training.
- Only a single architecture is tested. This weakens claims of generality.

**Questions:**

- Did you consider using non-adjacent frames (e.g., larger temporal spacing)? This would isolate long-range temporal cues from redundancy effects and help test the stated hypothesis more cleanly.
- I would suggest providing a more comprehensive discussion a potential mechanistic explanation for why more temporal data is worse.
- I strongly suggest including at least one experiment on revMD17 or an ablation that explicitly examines the role of simulation noise. This would reduce concerns that the observed degradation with $T=3$ is partly due to noise accumulation, and likely increase the signal in the reported results.
- Consider increasing sample counts or reporting more seeds, especially for Table 4 / Figure 2. The variability observed makes interpretation difficult.
- Please phrase some claims more cautiously. It may be more accurate to say that "for short-range deterministic MD data, two frames suffice and three may introduce harmful redundancy" rather than the broader "less is more" framing.
- Make explicit that the method assumes deterministic MD with fixed time-step and no stochastic thermostat; otherwise readers may overgeneralize. Please comment on practical relevance given that MD trajectories are typically sub-sampled to avoid strongly correlated training data.
- Results with a second architecture (e.g., EGNN, NequIP, MACE, PET) would strengthen the argument for generality, but I do not view this as essential for acceptance.
- Using $\mathbf{p}$ for positions is an unconventional choice for MD, as it is commonly used for momenta. $\mathbf{r}$ is the usual choice. I would suggest changing this notation.
- It may be interesting to consider a different/additional toy system that has simple governing equations, but chaotic dynamics, for example a double pendulum.

---

> ### Author Response · Authors · 2025-11-29
>
> We thank the reviewer for the constructive report and the positive assessment of the paper’s clarity, practicality, and intuition. We respond to each weakness and question in turn.
>
> > Weaknesses:
> > 1. **Statistical signal is weak… high variance…**
>
> Although we appreciate the reviewer's concern, we respectfully disagree on their inference on significance of our results. We reported on two datasets MD17 and ISO17. MD17 is a heavily studied and saturated benchmark, and our baseline model, Equiformer, is very strong. In this regime, average improvements of a few percent in force MAE can support promising ideas which can provide significant insights into the underlying phenomenon. The gains are more pronounced in other settings: on ISO17, FRAMES (T=2) reduces energy MAE by more than a factor of 5 in the within-distribution split and nearly 6x in the out-of-distribution split, compared to the same Equiformer baseline. We believe our results are more than marginal improvements.
>
> Also, we note that the variance is not high in real-world datasets including MD17 and ISO17, but high mainly in our synthetic spring-mass experiment. This variance is likely due the experiment setup and can improve with more diversified data.
>
> > 2. **Using original MD17 instead of revMD17… known noise issues…**
>
> We appreciate this concern. We did not use revMD17 because it is no longer a trajectory dataset: it is a ∼100k subsample per molecule that is subsequently re-relaxed. The temporal ordering and continuity needed for FRAMES are therefore not available. Our method relies on true MD trajectories to define the displacement target $\Delta p_t = p_{t+1}-p_t$ and to vary the history length T, which is not possible on revMD17.
>
> > 3. **Paper sometimes occasionally overstates…**
>
> Thank you for suggestion. We agree with the reviewer’s comment and have toned down our claims in the updated manuscript.
>
> > 4. **Method limited to deterministic, fixed-step MD…**
>
> Although, we see merit in the reviewer's comment, we believe it is not entirely true. Our method can be applied in stochastic environments as well, provided the underlying dynamics is still stochastically first-order. There is analogy that can be drawn from flow-matching models with our model. Simplistically, as flow-matching models can approximate the stochastic diffusion process with deterministic process, we can apply our deterministic method to stochastic first order methods, plausibly with reasonable performance. However, testing this is beyond the scope of current work but remains interesting future work. We thank the reviewer for taking the attention to this important aspect.
>
> > Questions:
> > 1. **Did you consider using non-adjacent frames? ...**
>
> Yes, we have used non-adjacent frames in our experiments. Adjacent frames led to weaker results, possibly because of noise. We will make this clearer in the experiments section.
>
> > 2. **Provide explanation for why more temporal data is worse.**
>
> We support our hypothesis with an explanation via synthetic spring-mass experiment. The experiment shows that when the underlying dynamics is of first-order, the trajectory can efficiently be modelled with $T=1$ or $T=2$. This provides some evidence to support our hypothesis that when we are able to extract information with few frames, it is likely that the underlying MD that generated the data is approximately of the first-order.
>
> > 4. **Add more seeds / samples on the spring-mass example.**
> > 5. **Phrase claims more cautiously (“for deterministic MD, two frames suffice …”).**
>
> Thank you for suggestion. We updated the results, and manuscript accordingly.
>
> > 6. **Make explicit that the method assumes deterministic MD ...**
>
> Thank you for your insightful comment. We have updated the manuscript to explicitly state that our approach assumes deterministic MD simulations with fixed steps and no stochastic thermostats. Regardless, we believe our method is extensible to stochastic dynamics with few approximations, which are beyond the scope of this work.
>
> Regarding the practical relevance of our approach, we acknowledge that while subsampling helps to reduce temporal correlations and provides a more statistically independent set of data points. Further investigation is needed to understand how subsampling might influence the results when applying our method to real-world data.
>
> > 7. **Trying a second architecture strengthens generality (optional).**
>
> Equiformer is a strong and widely used baseline, and due to limited computational resources we focused our experiments on this architecture. The FRAMES objective itself is architecture-agnostic: it only adds an auxiliary temporal loss on top of an existing force-prediction backbone and can in principle be applied to other MLIP architectures without modification. Subject to time and resources, we plan to include additional results with EquiformerV2 in the appendix.
>
> > 8. **Using $p$ for positions is an unconventional...**
>
> Sure, we have updated our notations.

---

### Official Review · Reviewer_iz1o · 2025-11-09

**Soundness:** 3
**Presentation:** 3
**Contribution:** 2
**Rating:** 6
**Confidence:** 2

**Summary:**

This paper introduces FRAMES, a novel training strategy for improving neural network predictions of molecular dynamics (MD). It proposes a novel training approach that uses an auxiliary loss function during training to leverage temporal information from MD trajectories. The backbone FRAMES model is the Equiformer (equivariant GNN). Through an additional auxiliary head on top of the traditional prediction head, it regularize the model training via atom position changes across frames $\Delta{p_t}$ from concatenated frame embeddings. In the experiments, the author demonstrates an interesting finding that minimal temporal information (2 consecutive frames) is optimal
Using 3+ frames introduces data redundancy and degrades performance - "Less is More". Across two real-world MD benchmark and one synthetic datasets, FRAMES(T=1) achieves better performance than Euiformer and (T>1) variants.

**Strengths:**

1. The paper introduces an effective auxiliary loss that predicts the atomic displacement across time frames. As a result, it significantly improves the result on ISO17, especially on energy prediction.

2. Problem formulation and literature review is crystally clear and includes most of the relevant work.

**Weaknesses:**

1. The performance improvement is only measured in one of the SOTA model - Equiformer. The generalization of FRAMES optimization is not demonstrated.

2. The novelty might be limited as the only innovation is on introducing an auxiliary loss, which is common in neural network molecular predcitions.

**Questions:**

1. When T>1, $\mathcal{L}_\text{aux}$ calculates the displacement across non-adjacent frames, I am wondering what if $\mathcal{L}_\text{aux}(T=2)$ sums $\mathcal{L}_\text{aux}(T=1)$ at T and T-1. Does it perform better?

2. Do you have any insightful or theoretical explaination on why T=1 works better?

3. What does the X-axis in Figure 2 represent? I didn't understand its correlations with Table 4.

---

> ### Author Response · Authors · 2025-11-29
>
> Thank you for your valuable feedback and insightful comments! Please see below for our responses to specific queries.
>
> > 1. **The performance improvement is only measured in one of the SOTA model - Equiformer. The generalization of FRAMES optimization is not demonstrated.**
>
>
> Equiformer is the standard model that is still near the state of the art scores. Due to limited computational resources, we conducted all experiments on Equiformer. We emphasize that the FRAMES objective itself is architecture-agnostic: it only adds an auxiliary temporal loss on top of an existing force-prediction backbone. In principle it can be applied to other MLIP architectures without modification. Subject to time and computational constraints during the rebuttal period, we plan to report a small set of additional experiments with EquiformerV2 and include them in the appendix (if they complete in time).
>
> > 2. **The novelty might be limited as the only innovation is on introducing an auxiliary loss, which is common in neural network molecular predcitions.**
>
> Respectfully, we disagree. Our main novelty is not limited to introducing auxillary loss, but lies in utilizing the trajectory information via a simple loss while also deriving the insight that full tragjectory introduces data redundancy and just two frames can reasonably capture most of the available information in the trajectory. While the auxiliary loss is the technical mechanism we employ, our core innovation is built upon a significant finding regarding the efficient utilization of Molecular Dynamics (MD) data.
>
> Specifically, our main contribution is the counter-intuitive insight that full trajectory information introduces data redundancy, which paradoxically degrades the predictor's performance. The crucial physical information - the necessary relationship for distilling priors — is optimally and most efficiently captured by the temporal relationship between just two consecutive MD frames.
>
> > 3. **When T>1, $\text{aux}(L_i)$ calculates the displacement across non-adjacent frames. I am wondering what  $\text{aux}(L_T=2)$ vs $\text{aux}(L_T=1)$ at T and T-1. Does it perform better?**
>
>
> We think there has been a misunderstanding. For $T={2,3}$ the auxillary loss predicts the difference between the current fame and the frame after that. In other words, our current implementation of $L_{\text{aux}(L_T)}$ **always** predicts the displacement between *adjacent* frames, i.e., $\Delta p_t = p_{t+1} - p_t$, as defined in Eq. (4).
>
> >  4. **Do you have any theoretical explanations or insights on why T=1 (T=2) works better?**
>
> Although, we do not have theoretical proofs to support our hypothesis, we support our hypothesis via synthetic spring-mass experiment. The experiment shows that when the underlying dynamics is of first-order, the trajectory can efficiently be modelled with $T=1$ or $T=2$. This provides some evidence to support our hypothesis that when we are able to extract information with few frames (like in our experiments), it is likely that the underlying molecular dynamics which generated the data is approximately of the first-order.
>
> > 5. **What is Figure 2. X-axis? Not clear how it correlates with Table 4.**
>
> Figure 2. X-axis represents the number of frames the model take as input to calculate the aux loss for it. Figure 2. and Table 4. show the accuracy of model on the spring-mass data. Figure 2. shows the accuracy of a Linear model while Table 4. shows the accuracy of an MLP model. We have updated the captions accordingly.

---

### Meta-Review · Area_Chair_cUzK · 2026-01-11

**Summary:**

The paper proposes an auxiliary-loss training strategy (FRAMES) to exploit minimal temporal information from molecular dynamics trajectories, and reports improvements over a strong Equiformer baseline on MD17/ISO17, with an intuitive spring–mass toy example. Reviewers generally found the idea simple and clearly presented, and noted that the approach is easy to implement and incurs no inference-time cost.

However, several core concerns remain unresolved. Empirical validation is limited to a single backbone, leaving the isolation of the proposed training effect from architecture-specific behavior insufficiently established. The statistical signal on MD17 is weak in parts, with potential confounding from dataset noise and limited robustness analysis. Key claims about “less is more” rely on heuristic explanations and restricted benchmarks, and broader practical relevance beyond MD-based datasets is not convincingly demonstrated. Related comparisons (e.g., alternative auxiliary objectives or modern architectures) remain incomplete.

Overall, reviewer opinions are mixed, with marginal scores and explicit expressions of reservation, and no reviewer provided a clear post-rebuttal signal of increased confidence. Given the interrupted process, this assessment reflects a conservative, best-effort synthesis based solely on the available reviews and discussion, without assuming any score changes.

**Reviewer Concerns:**

Reviewer iz1o:

Addressed:
• Clarified that the auxiliary loss predicts displacement between adjacent frames; Figure 2 x-axis clarified.

Partially addressed:
• Explanation for why T=2 works best remains heuristic (toy-system intuition only).

Outstanding:
• Generalization beyond Equiformer is not empirically demonstrated.

Reviewer acmh:

Addressed:
• Notation issues and deterministic-MD assumption clarified.

Partially addressed:
• Claims toned down; non-adjacent frame usage explained, but without clean isolating ablations.

Outstanding:
• Potential MD17 noise confounding and limited statistical robustness.
• Validation restricted to a single architecture.

Reviewer 8Ljb:

Addressed:
• Toy baseline clarified; MD vs relaxation text corrected.

Partially addressed:
• Noisy-node baseline added, but temporality-specific advantage not unequivocally established.

Outstanding:
• Weak MD17 gains and lack of broader empirical validation (e.g., other architectures).

Reviewer ZeyN:

Addressed:
• Related work and MD description corrected.

Partially addressed:
• Practical relevance reframed, but without new empirical evidence.

Outstanding:
• Limited significance due to reliance on MD-only datasets and single-architecture evaluation.

**Reviewer Scores:**

Reviewer iz1o:

Original score: 6

Likely post-rebuttal score: 6

Justification:
• No explicit reviewer signal; generality concern remains.

Reviewer acmh:

Original score: 6

Likely post-rebuttal score: 6

Justification:
• No explicit reviewer signal; noise/robustness and single-architecture issues persist.

Reviewer 8Ljb:

Original score: 2

Likely post-rebuttal score: 2

Justification:
• No explicit reviewer signal; core skepticism about significance and causality remains.

Reviewer ZeyN:

Original score: 4

Likely post-rebuttal score: 4

Justification:
• No explicit reviewer signal; concerns about practical impact and evaluation breadth remain.

---

### Decision · Program_Chairs · 2026-01-26

Reject